# Facile Synthesis of Quinolinecarboxylic Acid–Linked Covalent Organic Framework via One–Pot Reaction for Highly Efficient Removal of Water–Soluble Pollutants

**DOI:** 10.3390/molecules28093752

**Published:** 2023-04-27

**Authors:** Mingzhu Yang, Wenhua Ji

**Affiliations:** 1Key Laboratory for Applied Technology of Sophisticated Analytical Instruments of Shandong Province, Shandong Analysis and Test Center, Qilu University of Technology (Shandong Academy of Sciences), Jinan 250014, China; ymz274517@163.com; 2Key Laboratory for Natural Active Pharmaceutical Constituents Research in Universities of Shandong Province, School of Pharmaceutical Sciences, Qilu University of Technology (Shandong Academy of Sciences), Jinan 250014, China

**Keywords:** covalent organic framework, multicomponent reaction, quinoline skeleton, carboxyl group, environmental pollutants

## Abstract

To efficiently eliminate highly polar organic pollutants from water has always been a difficult issue, especially in the case of ultralow concentrations. Herein, we present the facile synthesis of quinolinecarboxylic acid-linked COF (QCA–COF) via the Doebner multicomponent reaction, possessing multifunction, high specific surface area, robust physicochemical stability, and excellent crystallinity. The marked feature lies in the quinolinyl and carboxyl functions incorporated simultaneously to QCA–COF in one step. The major *cis*–orientation of carboxyl arms in QCA–COF was speculated by powder X–ray diffraction and total energy analysis. QCA–COF demonstrates excellent adsorption capacity for water–soluble organic pollutants such as rhodamine B (255.7 mg/g), methylene blue (306.1 mg/g), gentamycin (338.1 mg/g), and 2,4–dichlorophenoxyacetic acid (294.1 mg/g) in water. The kinetic adsorptions fit the pseudo–second order model and their adsorption isotherms are Langmuir model. Remarkably, QCA–COF can capture the above four water–soluble organic pollutants from real water samples at *ppb* level with higher than 95% removal efficiencies and excellent recycling performance.

## 1. Introduction

Covalent organic frameworks (COFs) are a class of designable, porous, crystalline polymers with outstanding merits of tunable functionality and flexible topological connectivity [1,2,3,4,5]. The precise arrangement of building blocks and rigid backbone affords highly crystalline materials, making COFs more robust than amorphous organic polymers [6]. Structural functionalization in COFs can provide performance enhancements, including adsorption, host–guest interactions, and optical/electrical responses [7,8,9,10]. The flexible pore wall decoration for the affiliated groups or active sites in COFs offers potential exploration in their functionality [11], while the covalent linkages contribute to the high thermal and chemical stabilities of COFs. The efficient fabrication of functionalized COFs is one of the most vital synthetic challenges, which largely affects the expansion of their application fields.

An alternative strategy to convert active sites of COFs into versatile linkages is the post-synthetic modification of some linkers. Recently, it has been demonstrated that the imine–linked COFs can be transformed into quinoline, pyridine, secondary amine, secondary amide, thiazole, and oxazole–connected COFs [12,13,14,15,16,17]. In such a way, it is easy to generate new COFs or improve their physical and chemical features [18], but it is difficult to prevent the framework from collapsing. Furthermore, it suffers from tedious work–up and high energy and time consumption in most cases. Aiming to achieve the diversity, feasibility, and generality of synthetic methodology, recent advances that have improved the stability of COFs include the Knoevenagel reaction [19,20], Aldol condensation [21], nucleophilic aromatic substitution [22,23], and multicomponent reactions [24,25,26,27,28,29,30]. The multicomponent reactions delicately combine a series of irreversible/reversible covalent assemblies in one step to conveniently construct ideal annular scaffolds. To create the cyclic linkages, they show potential superiorities: (i) more diversity of COFs in functionality and structure can be facilely realized; (ii) more organic monomers break the structural restriction of the design and synthesis of frameworks. However, the delicate control over specific multicomponent reactions to produce ordered COF materials remains an arduous challenge.

The incorporation of functionalized side arms into COFs is capable of carrying out new applications. Different functional side arms, involving –SH, –NH_2_, –CO_2_H, –CONR–, –CO_2_R–, –COSR–, and chiral groups, can be introduced into COFs using post–synthetic modification with the click reaction [9,31,32,33] or the esterification reaction [34,35,36,37]. However, it is rather difficult for functionalizing COFs as they lack suitable reactive groups in most cases [34]. The tedious synthetic efforts and potential interactions during the process of COFs formation also limit the post–synthetic modification [38]. Potentially, the multicomponent reactions can open up a novel route for the introduction of multivariate functions into COFs. Unfortunately, a few recent works upon applying the multicomponent reactions to build COFs mainly focus on their frameworks, where the linkages formed without any side functional arms.

In view of the adsorption–based separation featuring quick removal, ease of operation, and cost–effectiveness, some COF materials have been utilized as adsorbents to remove toxic organic pollutants from water [39,40], but they are mostly limited to the water–insoluble organic pollutants, involving per– and poly–fluorinated alkyl substances (PFASs), bisphenol A, and nitrobenzene. Furthermore, a few COF adsorptions documented were only performed for the removal of pollutants at high concentrations, for instance, PFASs at 200 ng/mL [41], methylene blue (MB) at 8000 ng/mL [42], and rhodamine B (RhB) at 10,000 ng/mL [43]. To efficiently eliminate highly polar organic pollutants from water has always been a difficult issue, especially in the case of ultralow concentrations. Therefore, it is of significance to develop new COF adsorbents for the efficient removal of toxic organic pollutants at ultralow concentrations from water.

In this work, we develop a simple and efficient approach to construct a quinolinecarboxylic acid–linked COF, QCA–COF, based on the Doebner multicomponent reaction (Figure 1). QCA–COF was facilely prepared under solvothermal conditions from 1,3,5–tris(4–aminophenyl)benzene (TAPB), *p*–phthalaldehyde (PDA), and pyruvic acid using sulfamic acid as a catalyst. Note that all monomers are easily available on a large scale. More importantly, compared with the previous reports [27,28,29,30], sulfamic acid can greatly simplify the reaction process and shorten the reaction time from 72 h to 8 h. In view of its multiple binding sites involving quinoline N, aromatic rings, and carboxyl functions, adsorption applications of QCA–COF for ultralow concentration water-soluble organic contaminants were evaluated.

## 2. Results and Discussion

### 2.1. Synthesis and Characterization

The Doebner multicomponent reaction catalyzed with sulfamic acid is a powerful method for the synthesis of quinoline–4–carboxylic acid derivatives [44]. We initiated our studies by evaluating the model three–component reaction of aniline, benzaldehyde, and pyruvic acid with sulfamic acid as a catalyst (see Appendix A). The NMR spectra confirmed that the desirable model compound quinoline–4–carboxylic acid was successfully synthesized (Appendix A). Next, to obtain QCA–COF with good crystallinity, the reaction conditions involving solvents and temperatures have been optimized. After many designable trials, we have successfully achieved the right conditions to yield QCA–COF (Appendix A). Highly crystalline QCA–COF was obtained in a solution of 1,4–dioxane/*n*–butanol (*v/v* = 1:4) at 110 °C for 8 h.

The chemical structure of QCA–COF was identified by solid–state ^13^C NMR, FT–IR, elemental analysis, and XPS techniques. The formation of QCA–COF was well supported by the FT–IR spectra. As dedicated in Appendix A, the characteristic absorptions at 1551 and 1603 cm^−1^ indicate the existence of quinolyl skeletons. The typical peaks around 1707 cm^−1^ are ascribed to the carboxyl functions. The solid-state ^13^C NMR spectrum of QCA–COF identifies the carboxyl function at 168 ppm, and the quinolyl group at ca. 156, 138, and 123 ppm (Appendix A). These results are similar to those of the model compound. In the XPS of QCA–COF (Appendix A), the characteristic peaks at 531.78 eV, 398.58 eV, and 285.58 eV represent the carboxyl O, quinolyl N, and quinolyl C atoms, respectively. Elemental analysis data of QCA–COF (C, 73.25%; H, 3.65%; N, 5.49%) essentially matched the values (C, 76.92%; H, 3.44%; N, 5.98%) calculated for its corresponding formula [C_90_H_48_O_12_N_6_]_n_. The scanning electron microscope image of QCA–COF gives a similar spherical shape (Appendix A). All characterizations imply that the expected QCA–COF has been successfully prepared by the one–pot reaction. It should be noted that our method is of significance for constructing COF materials that possess functionalized side arms aimed to further expand their applications.

The porosity of QCA–COF was evaluated with the N_2_ adsorption–desorption test at 77 K. As shown in Figure 2a, the IV isotherm indicated that QCA–COF has the characteristic mesoporous structure. Its N_2_ adsorption quantity at 77 K is 527 cm^3^/g, and its Brunauer−Emmett−Teller (BET) surface area is 716 m^2^/g. The pore size distribution based on Barrett−Joyner−Halenda isotherms (Figure 2a, inserts) center at 3.07 nm, which matches well with the data of the simulated structure for QCA–COF (3.0 nm).

Thermogravimetric analysis certificated that QCA–COF is stable enough to 420 °C, featuring good thermal stability (Appendix A). Furthermore, it was treated under different harsh conditions for 72 h to examine the chemical stability. All FT–IR spectra of QCA–COF almost remain unchanged (Appendix A) after treatments, showing its remarkable chemical stability. N_2_ adsorption−desorption investigations proved its permanent porosity under harsh environments (Appendix A).

Theoretically, QCA–COF generated via the Doebner three-component reaction should mainly exist in two isomers: *cis*–form and *trans*-form (Appendix A), which originated from the different spatial configurations of its *p*–phenyl-bridged *bis*–quinolinecarboxylic acid linkers. The PXRD analysis of QCA–COF displays peaks at 2.84°, 4.86°, 5.68°, and 7.46°, corresponding to the facets of (100), (110), (200), and (210), respectively (Figure 2b). Materials Studio software was used to conduct the lattice modeling and Pawley refinement of QCA–COF, which outputs the most probable structure being its *cis*–form with the eclipsed AA stacking. The crystallographic information was shown in Appendix A. The Pawley refinements afforded optimized parameters, which provide agreement factors: *R_wp_* = 12.33%, *R_p_* = 9.71% for its *cis*-form, and *R_wp_* = 17.97%, *R_p_* = 13.17% for its *trans*–form. Obviously, the former is more consistent with the experimental data. Moreover, the total energy for its *cis*–form is 191 kcal/mol, which is lower than that for its *trans*–form (195 kcal/mol). Thus, the PXRD analysis confirmed that QCA–COF exists mainly in its *cis*–form rather than in its *trans*–form. Note that after being treated under harsh conditions, the PXRD patterns of QCA–COF almost retain the identical characteristic peaks (Appendix A), indicating no framework collapsing.

To offer insights into this one–pot synthesis mechanism of QCA–COF, the imine–linked TAPB–PDA–COF was prepared and the post–synthetic modification was carried out. Pyruvic acid and sulfamic acid were combined with the precursor TAPB–PDA–COF under the same conditions (1,4–dioxane/*n*–butanol, 110 °C, 8 h). However, the elemental analysis certificated that we have not obtained the desired QCA–COF. This implied that the mechanism may not involve the formation process of imine but *α*,*β*–unsaturated carbonyl compound yielded as the Doebner–von Miller reaction [45]. Hence, a plausible mechanism was proposed for this three–component synthesis of QCA–COF (Appendix A): (i) a condensation between PDA and pyruvic acid to *α*,*β*–unsaturated keto acid I occurs, where the I quickly converts into its *cis*–form driven by the stabilization of H–bonding formation with TAPB; (ii) and (iii) a conjugate addition happens through amine to attack the *cis*–form of I, creating intermediate II followed by cyclization and aromatization to produce intermediate III; (iv) III continues to repeat (ii) and (iii) steps in turn, yielding a *cis*–edge IV of COF; and then, the sequence from (ii) to (iv) was repeated until QCA–COF was finally formed. This H–bonding directed synthesis of COFs with *cis*–functional arms is quite interesting.

### 2.2. Pollutant Removal Evaluation

Our obtained QCA–COF can be qualified as potential adsorbents for removing organic pollutants from water due to their intrinsic acidic and basic functions along with different aromatic nuclei. Thus, QCA–COF is used to evaluate its removal function of highly water–soluble contaminants, involving 2,4–D, RhB, MB, and gentamycin from water. The analysis details of four contaminants are shown in Appendix A. Delightedly, we found that all four organic pollutants can be removed >98% within 5 min, 9 min, and 15 min at 20.0 ng/mL, 100.0 ng/mL, and 200.0 ng/mL (Figure 3a–c), respectively.

Remarkably, our results are much superior to that of COF powder described previously, which was removed at 70% after 2 h [42]. Moreover, control experiments show that QCA–COF achieves much higher removal efficiencies for four pollutants at 20.0 ng/mL than TAPB–PDA–COF, QCA–COP, activated carbon, and ion exchange resin (Figure 4). These results demonstrate that QCA–COF can efficiently clean up the water contaminated with ultralow concentration highly water-soluble organic pollutants in the independent experiment.

The isothermal adsorption of four contaminants by QCA–COF was carried out to evaluate the maximum adsorption capacity. The details are shown in Appendix A. The amount of four pollutants adsorbed on QCA–COF increases along with their initial concentrations. The experimental maximum adsorption capacities for 2,4–D, RhB, MB, and gentamycin are 294.1, 255.7, 306.1, and 338.1 mg/g, respectively. Fitting with the Langmuir equation, the highest monolayer adsorption capacities are calculated to be 303.0 mg/g (2,4–D), 263.2 mg/g (RhB), 312.5 mg/g (MB), and 344.8 mg/g (gentamycin). This good adsorption capacity may be ascribed to the multiple binding sites and high specific surface area of QCA–COF. As shown in Appendix A, all values of *n* for the Freundlich model are more than two, demonstrating the easy occurrence of adsorption.

### 2.3. Adsorption Mechanism

As QCA–COF has multiple binding sites, unique porous structural frameworks, and large conjugate systems, it can interact with the heterocycle, electron–donating, or nitrogen–containing groups of adsorbates through a wide variety of contacts, such as H–bonding, π–π stacking, as well as hydrophobic, electrostatic, and hydrophilic interactions. As shown in the charge density distribution under natural pH conditions (Figure 5), most areas of QCA–COF are neutral, except for quinoline N and carbonyl O, which are negatively charged. For MB, the N element at the end of the structure is positive, while for gentamycin and RhB, most areas of their structure are slightly positive. In 2,4–D, the terminal H atoms of its structure are positive. These results proclaimed that the electrostatic interaction is a key driving force for QCA–COF adsorbing these contaminants.

Based on the consideration of quantum mechanics, the behavior of materials can be predicted and calculated via the DFT method. Hence, the information of adsorbates/adsorbent–combined configurations was studied. For adsorbates, the top-site and pore–site of COF are the potential adsorption sites. According to the statistical results of annealing configurations, almost all adsorbates are trapped in the pore–size rather than top–size [46]. The adsorption energies for QCA–COF-adsorbed 2,4–D, RhB, MB, and gentamycin are −51.2, −44.4, −57.1, and −79.6 kcal/mol, respectively, which indicated that the adsorption ability of QCA–COF for gentamycin is superior to that for 2,4–D, RhB, and MB.

### 2.4. Reusability

The high cost for synthesis of COFs is one of the main limitations for their applications. Accordingly, the reusability of QCA–COF was studied to make it more economical. Fortunately, RhB, MB, and gentamycin captured by QCA–COF can be completely desorbed using HCl aqueous solution (1.0 M). However, 2,4–D should be desorbed using methanol due to its poor solubility in the acidic solution. As shown in Figure 6, QCA–COF almost maintains the same removal efficiencies even after six cycles. Furthermore, QCA–COF remains stable in these solvents.

### 2.5. Application of QCA–COF

The applicability of QCA–COF to remove four environmental pollutants from real water samples was validated. The results are summarized in Table 1. Above 95% removal efficiencies indicate that QCA–COF has an anti–interference ability and strong affinity in real water samples, including pond water, tap water, and industrial wastewater. It is noteworthy that low adsorption capacities for Na^+^ (0.9 mg/g), Ca^2+^ (2.4 mg/g), and Mg^2+^ (1.7 mg/g) imply that QCA–COF can be applied for the purification of drinking water.

## 3. Materials and Methods

### 3.1. Chemical Reagents

Benzaldehyde (99.5%), aniline (99.5%), pyruvic acid (98%), PDA (98%), rhodamine B (RhB, 99%), methylene blue (MB, 98%), gentamycin sulfate (98%), 2,4–dichlorophenoxyacetic acid (2,4–D, 98%), *o*–dichlorobenzene (*o*–DCB, 99%), and sulfamic acid (99.5%) were purchased from Aladdin (Shanghai, China). 1,3,5–Tris(4–aminophenyl)benzene (TAPB, 98%) was purchased from J&K Scientific Ltd. (Beijing, China). Mesitylene (98%), 1,4–dioxane (99%), *n*–butanol (99%), acetic acid (99%), acetone (99%), and tetrahydrofuran (THF, 99%) were purchased from Sinopharm Chemical Reagent Co., Ltd. (Shanghai, China). These chemicals were used without further purification. The chemical structures of four water-soluble organic contaminants are shown in Appendix A.

### 3.2. Characterization and Chromatographic Conditions

Powder X–ray diffraction (PXRD) data were measured by a D8 ADVANCE X–ray powder diffractometer (Rigaku, Tokyo, Japan) under Cu Kα radiation (*λ* = 1.5405 Å). The range was from 2*θ* = 2.0° up to 30°. Scanning electron microscope (SEM) images were recorded on a SWPRATM55 scanning electron microscope (Carl Zeiss, AG, Aalen, Germany). The FT–IR data were obtained ranging from 500 to 3700 cm^−1^ on a Nicolet 710 spectrometer (Waltham, MA, USA). The N_2_ adsorption–desorption experiments were performed on a Kubo–X1000 analyzer (Bjbuilder, Beijing, China). The Brunauer−Emmett−Teller (BET) method was applied to calculate the surface area and pore volume. The pore size distribution was calculated with the Barrett−Joyner−Halenda model. Thermogravimetric (TG) analysis was performed from room temperature to 850 °C on the STA 449F3–QMS403C system (Netzsch, Germany). The X–ray photoelectron spectroscopy (XPS) was conducted using an ESCALAB 250XI imaging electron spectrometer (Thermo, Waltham, MA, USA). A Vario EL III analyzer (Elementar, Germany) was used to perform the elemental analysis.

The liquid–state NMR was recorded on an Avance III HD 400 MHz spectrometer (Bruker Biospin AG, Switzerland) using tetramethylsilane (TMS) as an internal standard. Chemical shifts were reported in ppm. ^1^H NMR spectra were referenced to DMSO–*d*_6_ (2.50 ppm), and ^13^C NMR spectra were referenced to DMSO–*d*_6_ (39.5 ppm). All ^13^C NMR spectra were measured with complete proton decoupling. The solid–state ^13^C NMR spectra were measured on a JNM–ECZ600R NMR spectrometer (JEOL, Japan). Contaminants were analyzed using the Waters ACQUITY UHPLC system coupled to a Waters Xevo TQ–XS triple quadrupole mass spectrometer (Milford, USA).

The analysis of four contaminants was performed using a Waters Acquity UPLC coupled with Waters Xevo TQ–XS (Milford, USA). The separations were obtained using a Waters Acquity UPLC BEH C18 column (1.7 µm, 2.1 × 50 mm) with the mobile phase, as listed in Appendix A, pumped at a flow rate of 0.3 mL/min. The temperature of the autosampler and column were 10 °C and 40 °C, respectively. The injection volume was 5 μL.

The mass spectrometry experiment was working with ESI mode under multiple reaction monitoring (MRM) conditions. The MRM details are listed in Appendix A. The ESI source parameters were: capillary voltage, 0.5 kV; source temperature, 150 °C; desolvation temperature, 400 °C; desolvation gas, 800 L h^−1^; desolvation gas, 99.99% purity of N_2_; and nebulization flow, 7.0 Bar.

The linear ranges and limits of quantification (LOQs) for the determination of four contaminants using UHPLC–MS/MS are listed in Appendix A.

### 3.3. Synthesis of QCA–COF

To a 10 mL round–bottom flask, PDA (60.3 mg, 0.45 mmol), TAPB (105.3 mg, 0.3 mmol), and 1,4–dioxane/*n*–butanol solution (5 mL, *v/v* = 1:4) were added. After they were sonicated for 10 min, pyruvic acid (79.2 mg, 0.9 mmol) and sulfamic acid (1 mol%) were added. The mixture was heated at 110 °C for 8 h and then cooled to room temperature. The yellow precipitate was collected by centrifugation and washed with water and THF, respectively. After Soxhlet extraction in acetone and THF for 6 h and drying under vacuum at 60 °C for 2 h, QCA–COF was obtained as a yellow powder in 53% isolated yield.

### 3.4. Synthesis of TAPB–PDA–COF

TAPB–PDA–COF was prepared using the procedure reported [47]. A mixture of TAPB (84.0 mg, 0.24 mmol), PDA (48.3 mg, 0.36 mmol), and acetic acid aqueous solution (6 M, 0.3 mL) in *o*–dichlorobenzene/*n*–butanol (3 mL, *v/v* = 1:1) was added in a Pyrex tube (35 mL) using three freeze–pump–thaw cycles. The tube was sealed and heated at 120 °C for 72 h. The precipitate was collected by centrifugation, washed with THF, and dried at 120 °C under vacuum overnight to give TAPB–PDA–COF.

### 3.5. Organic Contaminants Removal Experiments

QCA–COF (10 mg) was soaked in 20 mL of four aqueous solutions containing RhB, MB, 2,4–D, and gentamycin at three concentrations (20.0 ng/mL, 100.0 ng/mL, and 200.0 ng/mL), respectively. Each mixture was stirred at 22 °C. The concentrations of the supernatant solutions were analyzed through the ultrahigh performance liquid chromatography tandem mass spectrometry (UHPLC–MS/MS) at different times. All experiments were performed in triplicate. The removal efficiencies (R, %) of four contaminants at different times were calculated as follows:R (%) = (*C_i_ − C_t_*)/*C_i_* × 100%
where *C_i_* (ng/mL) is the initial concentration in solution, and *C_t_* (ng/mL) is the contaminant concentration in the system at time *t* (min).

Control experiments were performed using TAPB–PDA–COF without modification, and QCA–COP without crystallinity, activated carbon, and ion exchange resin.

### 3.6. Computational Method

The DMol3 program was used to perform all the spin-polarized density functional theory (DFT) calculations within the generalized gradient approximation (GGA) using the Perdew–Burke–Ernzerhof (PBE) formulation. The projected augmented wave potentials were chosen to describe the ionic cores. Based on a plane wave basis set, the valence electrons were taken into account. DFT semi-core pseudopotential was used for the core-electron treatment. The SCF convergence for each electronic energy was set as 1.0 × 10^−5^ Ha. The geometry optimization convergence criteria were set up as follows: 1.0 × 10^−5^ Ha for energy, 0.004 Ha Å^−1^ for force, and 0.01 Å for displacement, respectively. In addition, the van der Waals interactions have been considered in the calculation.

Under the environment of water, the surface charge distribution for four contaminants, QCA–COF, and TAPB–PDA–COF were simulated using the DFT method with the DMol3 program of Materials Studio 8.0. Double numerical basis with polarization functions was selected as the basis set. GGA–PBE was selected as the exchange–correlation functional. Grimme dispersion correction was used in all calculations to describe *π*-stacking and van der Waals interactions.

## 4. Conclusions

A powerful one–step synthetic strategy for constructing carboxy–functionalized quinoline–linked COFs has been successfully developed. Multivariate functions are smartly incorporated into the robust COF, which largely avoids the tedious work-up of post-synthetic modifications. Our QCA–COF possesses carboxyl-functionalized side arms and inherent physicochemical stability, so it can be qualified as an outstanding absorbent material. Compared with the reported sorbents, at ultralow concentrations (20–200 ng/mL) the water–soluble organic contaminants can be effectively purged from water. We anticipate that our robustly synthetic strategy may be utilized to fabricate a great variety of functionalized quinoline-bridged COF materials that show a wider range of applications in the future.

## Figures and Tables

**Figure 1 molecules-28-03752-f001:**
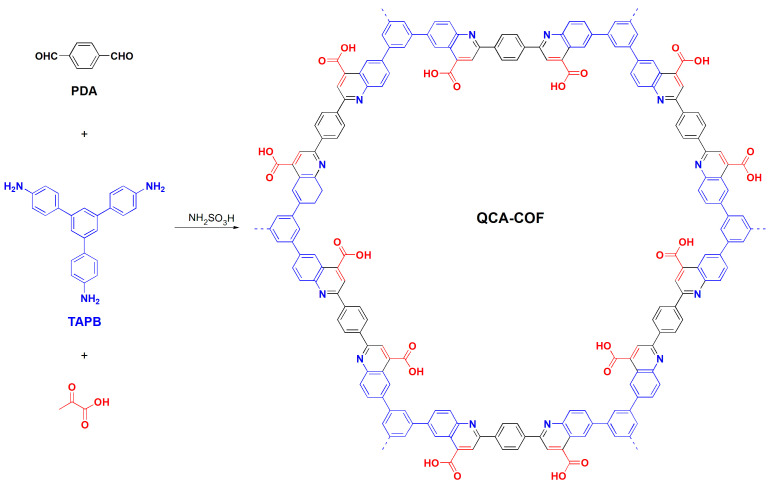
Synthesis of QCA–COF via the Doebner three–component reaction.

**Figure 2 molecules-28-03752-f002:**
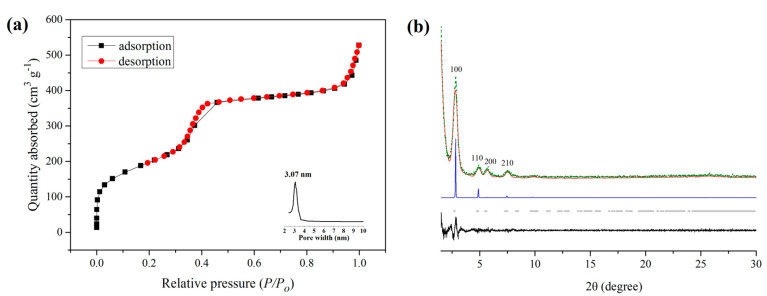
The nitrogen adsorption–desorption isotherms and pore size distribution for QCA–COF (**a**) and PXRD patterns of QCA–COF (**b**) observed experimentally (red), Pawley–refined patterns of its *cis*–form (green), difference between the experimental and calculated data (black), calculated patterns of its *cis*-form for AA (blue), and Bragg positions (gray).

**Figure 3 molecules-28-03752-f003:**
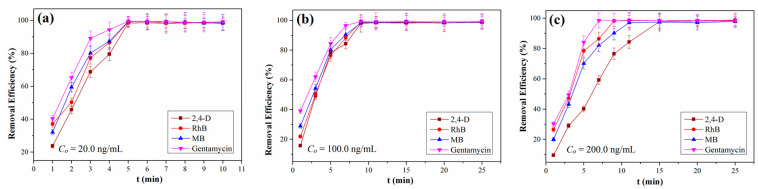
The removal efficiencies of QCA–COF at the initial concentration of 20.0 ng/mL (**a**), 100.0 ng/mL (**b**), and 200.0 ng/mL (**c**).

**Figure 4 molecules-28-03752-f004:**
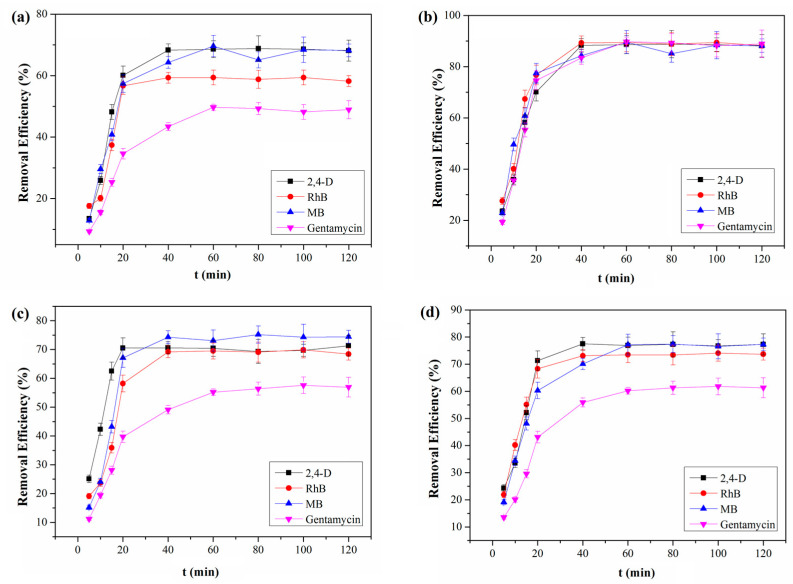
The removal efficiencies of TAPB–PDA–COF (**a**), QCA–COP (**b**), activated carbon (**c**), and ion exchange resin (**d**) for four water-soluble organic pollutants at 20.0 ng/mL.

**Figure 5 molecules-28-03752-f005:**
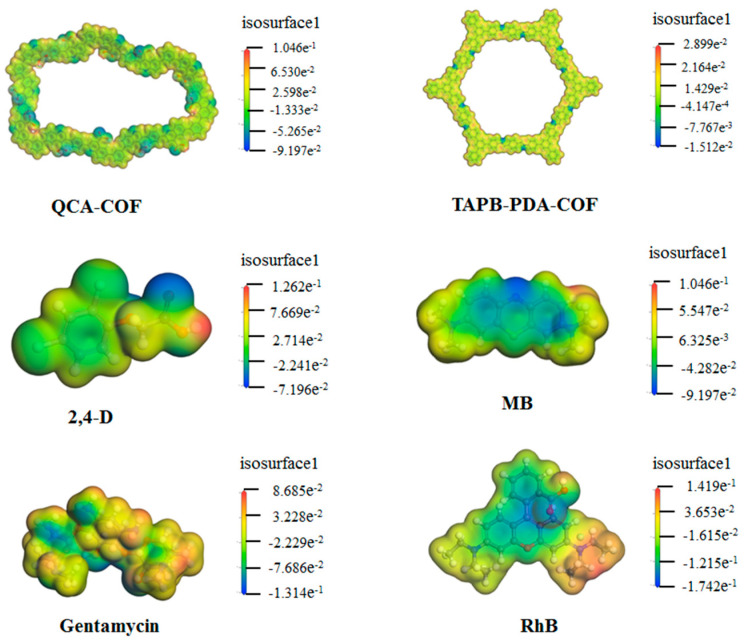
The charge distributions of QCA–COF, TAPB–PDA–COF, 2,4–D, gentamycin, MB, and RhB under the environment of water.

**Figure 6 molecules-28-03752-f006:**
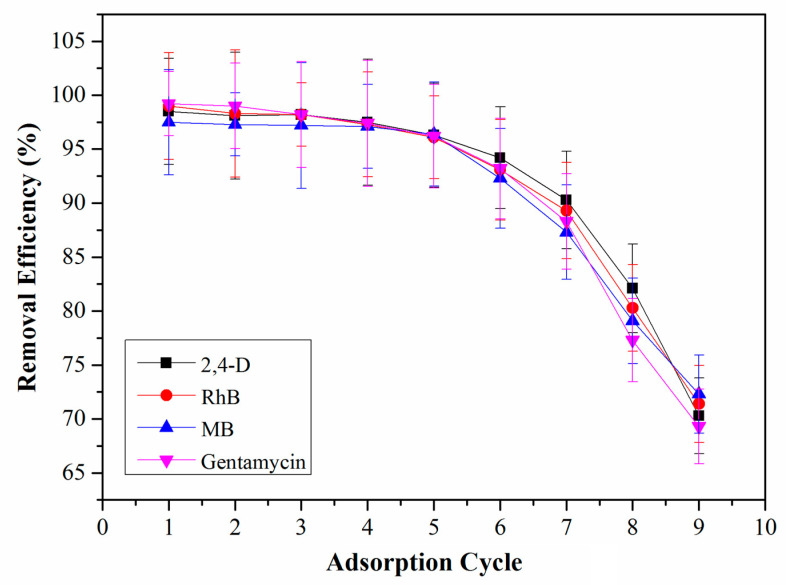
The regeneration performance of QCA–COF.

**Table 1 molecules-28-03752-t001:** The removal efficiency of QCA–COF for tap water, pond water, and industrial wastewater ^[a]^.

	Pond Water ^[b]^			Tap Water ^[b]^			Industrial Wastewater ^[b]^		
	Found/Spiked (ng/mL)	After Removal (ng/mL)	Removal (%)	Found/Spiked (ng/mL)	After Removal (ng/mL)	Removal (%)	Found/Spiked (ng/mL)	After Removal (ng/mL)	Removal (%)
RhB	ND ^[c]^	ND ^[c]^	-	ND ^[c]^	ND ^[c]^	-	ND ^[c]^	ND ^[c]^	-
	10.0 ^[e]^	0.18	98.2	10.0 ^[e]^	0.32	96.8	10.0 ^[e]^	0.31	96.9
	40.0 ^[e]^	0.54	98.6	40.0 ^[e]^	1.47	96.3	40.0 ^[e]^	1.35	96.6
	80.0 ^[e]^	1.32	98.4	80.0 ^[e]^	2.86	96.4	80.0 ^[e]^	3.25	95.9
MB	ND ^[c]^	ND ^[c]^	-	ND ^[c]^	ND ^[c]^	-	ND ^[c]^	ND ^[c]^	-
	10.0 ^[e]^	0.39	96.1	10.0 ^[e]^	0.23	97.7	10.0 ^[e]^	0.28	97.2
	40.0 ^[e]^	0.65	98.4	40.0 ^[e]^	1.59	96.0	40.0 ^[e]^	1.94	95.2
	80.0 ^[e]^	1.44	98.2	80.0 ^[e]^	3.38	95.8	80.0 ^[e]^	3.24	96.0
Gentamycin	ND ^[c]^	ND ^[c]^	-	7.72 ^[d]^	0.25	96.7	ND ^[c]^	ND ^[c]^	-
	10.0 ^[e]^	0.23	97.7	10.0 ^[e]^	0.76	95.7	10.0 ^[e]^	0.21	97.9
	40.0 ^[e]^	0.73	98.2	40.0 ^[e]^	1.23	97.4	40.0 ^[e]^	1.48	96.3
	80.0 ^[e]^	2.01	97.5	80.0 ^[e]^	3.76	95.7	80.0 ^[e]^	2.41	97.0
2,4-D	ND ^[c]^	ND ^[c]^	-	ND ^[c]^	ND ^[c]^	-	14.7 ^[d]^	0.29	98.0
	10.0 ^[e]^	0.21	97.9	10.0 ^[e]^	0.49	95.1	10.0 ^[e]^	0.59	97.6
	40.0 ^[e]^	0.89	97.8	40.0 ^[e]^	1.13	97.1	40.0 ^[e]^	2.63	95.2
	80.0 ^[e]^	2.97	96.3	80.0 ^[e]^	3.62	95.5	80.0 ^[e]^	3.74	96.0

^[a]^ Removal conditions as follows: sample volume, 25 mL; QCA–COF, 10 mg; adsorption temperature, 25 °C; adsorption time, 30 min. ^[b]^ Tap water was collected from our laboratory (Jinan, China), pond water came from the Shandong Analysis and Test Center (Jinan, China), and industrial wastewater was collected from a pharmaceutical factory (Jinan, China). ^[c]^ Not detected. ^[d]^ Found concentration. ^[e]^ Spiked concentration.

## Data Availability

Data are contained within the article.

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
