# Peer review of "Facile Synthesis of Quinolinecarboxylic Acid–Linked Covalent Organic Framework via One–Pot Reaction for Highly Efficient Removal of Water–Soluble Pollutants"

_molecules, 2023, doi:10.3390/molecules28093752_

Round 1
Reviewer 1 Report
The manuscript entitled “Facile Synthesis of Quinolinecarboxylic Acid-linked Covalent Organic Framework via One-pot Reaction for Highly Efficient Removal of Water-soluble Pollutants” by Ji et al describes the facile synthesis of quinolinecarboxylic acid-linked COF (QCA-COF) via the Doebner multicomponent reaction and the excellent adsorption capacity of QCA-COF for water-soluble organic pollutants. It is impressive that the water-soluble organic contaminants at ultralow concentrations (20-200 ng/mL) can be effectively purged from water. Therefore, the manuscript can be accepted for publication in Molecules after the following points being addressed.
1) To my knowledge, there are four relevant papers on quinolinecarboxylic acid-linked COF synthesis that should be cited. Chem. Commun. 2022, 58, 2508; J. Mater. Chem. A 2022, 10, 3346; Nat. Commun. 2022, 13, 2615; J. Am. Chem. Soc. 2023, 145, 2975.
2) A methodological description of powder X-ray diffraction refinement is lacking in the experimental section.
3) The Bragg positions should be given in the refinement results graph (Fig. 2b).
4) Lack of the space group information. From Table S2, it appears that the authors should have used a triclinic system. In general, hcb-type COFs crystallize in the hexagonal system.
5) Some papers on the facile synthesis of COFs and their water treatment applications should be cited. For an example, Molecules 2022, 27, 8002.
Author Response
(1) To my knowledge, there are four relevant papers on quinolinecarboxylic acid-linked COF synthesis that should be cited. Chem. Commun. 2022, 58, 2508; J. Mater. Chem. A 2022, 10, 3346; Nat. Commun. 2022, 13, 2615; J. Am. Chem. Soc. 2023, 145, 2975.
Re: Four relevant papers on quinolinecarboxylic acid-linked COF synthesis have been cited. Please see references 27-30.
(2) A methodological description of powder X-ray diffraction refinement is lacking in the experimental section.
Re: The methodological description of powder X-ray diffraction refinement. Please see the section 3.2 “…Powder X-ray diffraction (PXRD) data were measured by a D8 ADVANCE X-ray powder diffractometer (Rigaku, Tokyo, Japan) under Cu Kα radiation (λ = 1.5405 Å). The range was from 2θ = 2.0° up to 30°….”
(3) The Bragg positions should be given in the refinement results graph (Fig. 2b).
Re: The Bragg positions have been given in the refinement results graph. Please see Fig. 2b.
(4) Lack of the space group information. From Table S2, it appears that the authors should have used a triclinic system. In general, hcb-type COFs crystallize in the hexagonal system.
Re: The space group information has been added in the Table S2.
(5) Some papers on the facile synthesis of COFs and their water treatment applications should be cited. For an example, Molecules 2022, 27, 8002.
Re: The paper on the facile synthesis of COFs and their water treatment applications has been cited. Please see the reference 40.

Reviewer 2 Report
The manuscript “Facile synthesis of Quinolinecarboxylic Acid-linked Covalent Organic Framework via One-pot Reaction for Highly Efficient Removal of Water-soluble Pollutants” reports a new COF via the Doebner multicomponent reaction of 1,3,5-tris(4-aminophenyl)benzene, p-phthalaldehyde, and pyruvic acid. The developed COF possesses multifunction, high specific surface area, robust physicochemical stability, and excellent crystallinity. The experiments were well designed, and the products were well characterized. Actually, the present paper just expands the methodology developed in doi: 10.1039/D1CC06461D. In my opinion, the developed COF could be useful also for metal ions extraction. I recommend the manuscript for acceptance after minor revision.
1. For the synthesis of a model compound (quinoline-4-carboxylic acid) by a model three-component reaction of aniline, benzaldehyde, and pyruvic acid with sulfamic acid as a catalyst, the yield and the reaction scheme (including reaction conditions) should be given. Or it would be much better if the authors provided synthetic procedure, isolation procedure, scheme and the yield for this compound in SI or in M&M section.
2. Table S1: ratio of what is given?
3. It would be nice, if the authors give some data in the introduction on the reported earlier QCA-COFs (e.g. doi: 10.1039/D1CC06461D) to show the novelty.
Author Response
1. For the synthesis of a model compound (quinoline-4-carboxylic acid) by a model three-component reaction of aniline, benzaldehyde, and pyruvic acid with sulfamic acid as a catalyst, the yield and the reaction scheme (including reaction conditions) should be given. Or it would be much better if the authors provided synthetic procedure, isolation procedure, scheme and the yield for this compound in SI or in M&M section.
Re: The synthesis of quinoline-4-carboxylic acid has been given in SI. Please see “…A mixture of pyruvic acid (106 mg, 1.2 mmol), aniline (102 mg, 1.1 mmol), and benzaldehyde (106 mg, 1.0 mmol) in 1,4-dioxane/n-butanol solution (5 mL, v/v = 1:4) was well stirred with NH2SO3H (3 mg, 3 mol%). This mixture was magnetically stirred under 100 oC for 4h. Then, the reaction mixture was cooled to room temperature and concentrated to dryness in vacuo. The residue was recrystallized using ethanol to obtain quinoline-4-carboxylic acid as white solid (189 mg, 76%). 1H NMR (400 MHz, d6-DMSO): δ 14.03 (s, 1H), 8.69 (d, 1H, J = 8.0 Hz), 8.49 (s, 1H), 8.32 (d, 1H, J = 8.0 Hz), 8.19 (d, 2H, J = 8.0 Hz), 7.88 (t, 1H, J = 8.0 Hz), 7.73 (t, 1H, J = 7.2 Hz), 7.53-7.63 (m, 3H)…”.
2. Table S1: ratio of what is given?
Re: The ratio has been revised in the Table S1.
3. It would be nice, if the authors give some data in the introduction on the reported earlier QCA-COFs (e.g. doi: 10.1039/D1CC06461D) to show the novelty.
Re: The last paragraph of introduction has been revised. Please see “…More importantly, comparing with the previous reports [27-30], sulfamic acid can greatly simplifies the reaction process and shortens the reaction time from 72 h to 8 h…”.
